EMBO
Molecular Medicine

# Promoterless gene targeting without nucleases rescues lethality of a Crigler-Najjar syndrome mouse model

Fabiola Porro[1,‡], Giulia Bortolussi[1,‡], Adi Barzel[2,†], Alessia De Caneva[1], Alessandra Iaconcig[1], Simone Vodret[1], Lorena Zentilin[1], Mark A Kay[2] & Andrés F Muro[1,*] iD

## Abstract

Crigler-Najjar syndrome type I (CNSI) is a rare monogenic disease characterized by severe neonatal unconjugated hyperbilirubinemia with a lifelong risk of neurological damage and death. Liver transplantation is the only curative option, which has several limitations and risks. We applied an *in vivo* gene targeting approach based on the insertion, without the use of nucleases, of a promoterless therapeutic cDNA into the albumin locus of a mouse model reproducing all major features of CNSI. Neonatal transduction with the donor vector resulted in the complete rescue from neonatal lethality, with a therapeutic reduction in plasma bilirubin lasting for at least 12 months, the latest time point analyzed. Mutant mice, which expressed about 5–6% of WT Ugt1a1 levels, showed normal liver histology and motor-coordination abilities, suggesting no functional liver or brain abnormalities. These results proved that the promoterless gene therapy is applicable for CNSI, providing therapeutic levels of an intracellular ER membrane-bound enzyme responsible for a lethal liver metabolic disease.

**Keywords** brain damage; genetic liver disease; homologous recombination; hyperbilirubinemia; kernicterus

**Subject Categories** Genetics, Gene Therapy & Genetic Disease

## Introduction

The Crigler-Najjar syndrome type I (CNSI) is a rare monogenic pediatric disease (0.6–1 cases per $10^6$ live births) caused by a deficiency in the liver-specific uridine diphosphate glucuronosyltransferase 1A1 (*UGT1A1*), resulting in severe unconjugated hyperbilirubinemia since birth, with lifelong risk of permanent neurological damage, kernicterus, and death (Crigler & Najjar, 1952; Huang *et al*, 1970).

Current clinical practice consists of intense phototherapy (PT) treatment (12–14 h/day), but it becomes less effective with age, leaving liver transplantation as the only therapeutic option, with all the limitations and risks of the approach (Adam *et al*, 2012; Fagiuoli *et al*, 2013).

Gene replacement mediated by adeno-associated virus (AAV) is a promising approach (Kay, 2011; Mingozzi & High, 2011; Nathwani *et al*, 2011). However, loss of episomal DNA and therapeutic efficacy may result from hepatocyte duplication, which is very prominent in the neonatal/pediatric period (Cunningham *et al*, 2008; Wang *et al*, 2012; Bortolussi *et al*, 2014b).

Gene editing with sequence-specific endonucleases (Urnov *et al*, 2010; Joung & Sander, 2013; Wang *et al*, 2016) results in the permanent correction of disease-causing mutations. However, these approaches face a number of significant adverse effects, such as immunogenicity of the endonucleases, off-target cleavage and mutagenesis, and induction of chromosomal aberrations, concerns which are enhanced by the long-term expression of the nucleases (Carroll, 2014). Moreover, off-target integration of the transgene and endonuclease vectors bearing potent gene promoters carries the potential of transactivating cancer-related genes (Fu *et al*, 2013; Chandler *et al*, 2015).

The GeneRide strategy (Barzel *et al*, 2015), based on the targeted insertion of a promoterless therapeutic cDNA into the albumin locus without the use of nucleases, overcomes most of the safety concerns of gene therapy and holds several advantages: (i) site-specific integration with transgene regulation and transcription controlled by the robust liver-specific albumin promoter (Tilghman & Belayew, 1982; Pinkert *et al*, 1987); (ii) lack of a promoter in the therapeutic vector, thus reducing the probability of activating neighboring oncogenes by random integration (Donsante *et al*, 2007); and (iii) no risks of gene inactivation by nuclease-mediated off-target, transactivation, and hepatocellular carcinoma (HCC) development by insertion of the nucleases coding vectors.

In the present work, we have successfully applied the promoterless approach without nucleases to a relevant lethal mouse model of the CNSI.

1 International Centre for Genetic Engineering and Biotechnology (ICGEB), Trieste, Italy
2 Departments of Pediatrics and Genetics, Stanford University, Stanford, CA, USA
 *Corresponding author. Tel: +39 040 3757369; Fax: +39 040 226555; E-mail: muro@icgeb.org
 †Present address: Department of Biochemistry and Molecular Biology, The George S. Wise Faculty of Life Sciences, Tel Aviv University, Tel Aviv, Israel
 ‡These authors contributed equally to this work as first authors

# Results

### Targeting the albumin locus without nucleases results in precise integration of the eGFP cDNA

We first tested the promoterless gene editing approach without nucleases (Barzel *et al*, 2015) in WT animals by transducing a donor construct bearing the eGFP cDNA (rAAV8-Alb-eGFP), which contained the eGFP cDNA preceded by the 2A-peptide and flanked by the albumin homology arms (Fig 1A). Spontaneous site-specific recombination results in a fused mRNA (albumin-P2A-eGFP) transcribed by the strong albumin promoter that, due to the peptide 2A, is translated into two separate proteins.

At post-natal day 2 (P2), P4, P10, and P30 WT pups were transduced with rAAV8-Alb-eGFP (0.7E12 and 1.0E12 vgp/mouse) and sacrificed at P30 (I.P., P2, P4, and P10) and at P45 (i.v., P30) (Fig EV1A). Histological analysis of liver sections showed that the highest recombination rate was obtained in P4 pups transduced with the higher AAV dose, reaching about 0.14% of hepatocytes (Figs 1B and C, and EV1B and EV2A), with a marked decrease in those injected at P10 and P30. AAV-mediated episomal expression was used for comparative purposes (Fig 1B).

RT–PCR analysis, using primers specific for the recombinant hybrid mRNA, resulted in the expected product (Fig EV1C–E). Expression of the chimeric mRNA, determined by qRT–PCR, correlated with recombination efficiency (Fig 1D).

At 4 months after neonatal transduction, the proportion of eGFP-positive hepatocytes was roughly the same to that observed at P30 in the P4-high-dose group (Fig EV2A), while it was significantly reduced in the other groups. Semi-quantitative RT–PCR analysis of 4-month-old treated mice showed higher levels of PCR product in the mice injected at P4, as compared with samples from mice injected at P2 (Fig EV1D).

Interestingly, when 1.0E12 vgp/mouse, from the same AAV batch preparation, was injected as a single dose, or divided in three injections performed the same day (P4), separated by 5–6 h, the efficiency of multiple injections (m.i.) was much higher than the single one. Even the highest dose transduced in a single injection (4.0E12 vgp/mouse) resulted in less eGFP-positive cells than 1.0E12 given as m.i. (Fig EV2B and C).

### hUgt1a1 integration into the albumin locus of Ugt1$^{-/-}$ mice results in a reduction in plasma bilirubin to therapeutic levels

Following the successful targeting of the eGFP reporter construct, we transduced a construct containing the human WT Ugt1a1 cDNA (rAAV8-Alb-hUgt1a1) to Ugt1$^{-/-}$ mutant mice.

These mice have a targeted mutation in the Ugt1 gene, complete absence of glucuronidation activity, severe hyperbilirubinemia from birth, severe cerebellar abnormalities, and death by kernicterus before P16 [(Bortolussi *et al*, 2012, 2014a) and Fig EV3A–C]. On the contrary, PT treatment from birth up to P15 results in survival of all mutant mice (Bortolussi *et al*, 2014a; Fig EV3B).

Therefore, to demonstrate the feasibility and efficacy of the protocol, we performed our first experiment in non-lethal conditions. Mice were injected with AAV8-Alb-hUgt1a1 at P2, P4, and P10, at two doses (0.7E12 and 1.0E12 vgp/mouse), using the multiple injections procedure. Mice were treated with 15 days of PT from birth and sacrificed at P30 (Fig EV3D).

All transduced mice showed a significant reduction in plasma bilirubin levels (Fig EV3E), compared with controls (non-AAV transduced mutant mice, treated with PT up to P15), with values well below those causing neurological damage (Bortolussi *et al*, 2014a). Mice transduced at P4 with the highest AAV dose (1.0E12 vgp/mouse) showed the lowest plasma bilirubin levels, confirming the results obtained with the AAV8-Alb-eGFP vector (Fig 1B–D). The decrease in plasma bilirubin levels indicates that the hUgt1a1 enzyme, derived from a chimeric mRNA, was functional.

### Lethality of mutant mice is rescued by hUgt1a1 gene targeting into the albumin locus

Finally, to determine the therapeutic potential of the approach, we tested the procedure in lethal conditions, by treating mutant mice with PT for only 8 days after birth. This shorter PT treatment results in death of almost all mutant mice before P20 [7% survival (Bortolussi *et al*, 2014a)] but, at the same time, presents a "safe" time-window in which homologous recombination and transgene expression may occur after viral transduction.

We transduced mutant mice at P4 with AAV8-Alb-hUgt1a1 (1.0E12 vgp/mouse) and maintained the animals under PT up to P8 (Fig 2A). The treatment resulted in survival of all treated mutant mice while all untreated mutant mice died (Fig 2B). Monitoring of the animals showed no obvious abnormalities up to 12 months, the last time point examined. AAV8-Alb-hUgt1a1-treated mice showed low levels of plasma bilirubin, with an increase in bilirubin levels during the first months, while it increased rapidly and constantly till death in untreated animals (Fig 2C), as previously shown (Bortolussi *et al*, 2014a). In the following months, total bilirubin values remained stable in all AAV8-Alb-hUgt1a1-treated mice. Despite bilirubin concentration in

**Figure 1.  Transduction of WT neonate mice with the AAV-Alb-eGFP donor vector.**

A   Vector design and experimental scheme. Recombination of the AAV8 vectors (containing the eGFP or the WT human Ugt1a1 cDNAs, preceded by the 2A-peptide, and flanked by albumin gene homology arms) results in a fused "chimeric bicistronic" mRNA, which is translated into two separate proteins. Rectangles represent exons; thick black lines, introns and intergenic regions; thin gray lines, extragenic DNA sequences. Modified from Barzel *et al* (2015).

B   Histological analysis of liver sections. P2, P4, and P10 WT mice were IP transduced with rAAV8-Alb-eGFP (0.7E12 and 1E12 vgp/mouse) and sacrificed at P30. P30 WT mice were i.v. transduced and sacrificed at P45. As controls, WT mice were transduced with an episomal AAV8 vector (Bortolussi *et al*, 2014b). Nuclei were counterstained with Hoechst. Each field corresponds to a single animal ($n = 3$ per time point/treatment). Scale bar, 400 μm.

C   Quantification of the number of eGFP-positive hepatocytes. Two-way ANOVA: interaction, $P = 0.0011$; dose, $P < 0.0001$; time of injection, $P < 0.0001$; Bonferroni *post hoc* tests (0.7E12 versus 1.0E12): P2, $t = 5.271$, $P < 0.001$; P4, $t = 8.755$, $P < 0.001$; P10, $t = 2.141$, $P =$ NS. P2 1.0E12 versus P4 1.0E12, $t = 4.226$, $P < 0.01$; $n = 3$ per group; #, not performed.

D   Quantitative RT–PCR of liver total RNA samples from panel (B). Each dot represents a single animal. Two-way ANOVA: interaction, $P = 0.0009$; dose, $P < 0.0001$; time of injection, $P = 0.0013$; Bonferroni *post hoc* tests (0.7E12 versus 1.0E12): P2, $t = 2.244$, $P =$ ns; P4, $t = 8.273$, $P < 0.001$; P10, $t = 1.647$, $P =$ ns; $n = 3$ per group.

Data information: Results are expressed as mean ± SD.

plasma of treated mutant animals was higher than WT (about 0.5 mg/dl), their levels were far from the toxic threshold during the whole experimental period.

To rule out the presence of functional abnormalities, we performed a motor-coordination test in a rotarod. Treated mutant mice performed similarly to WT untreated and WT-AAV-Alb-eGFP-treated

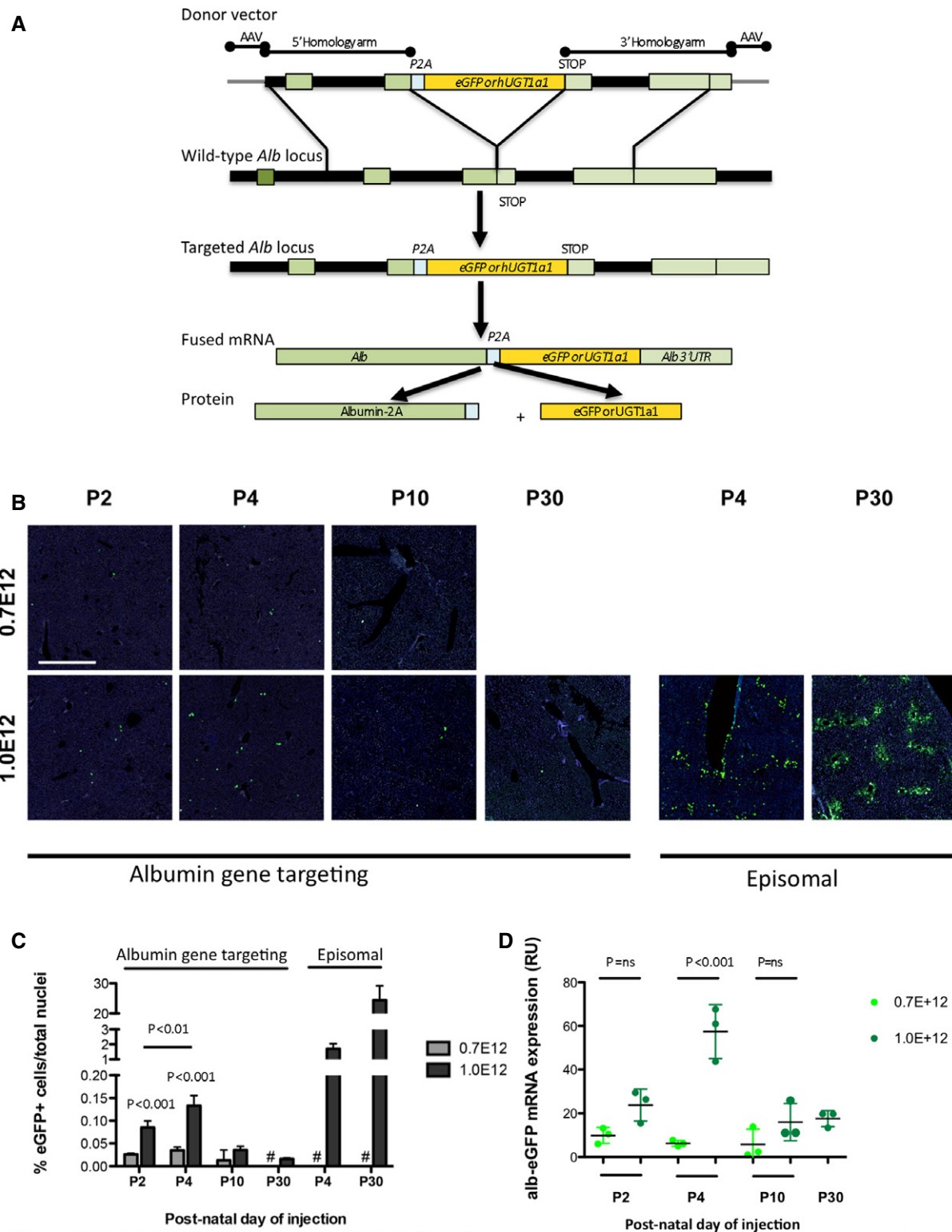

Figure 1.

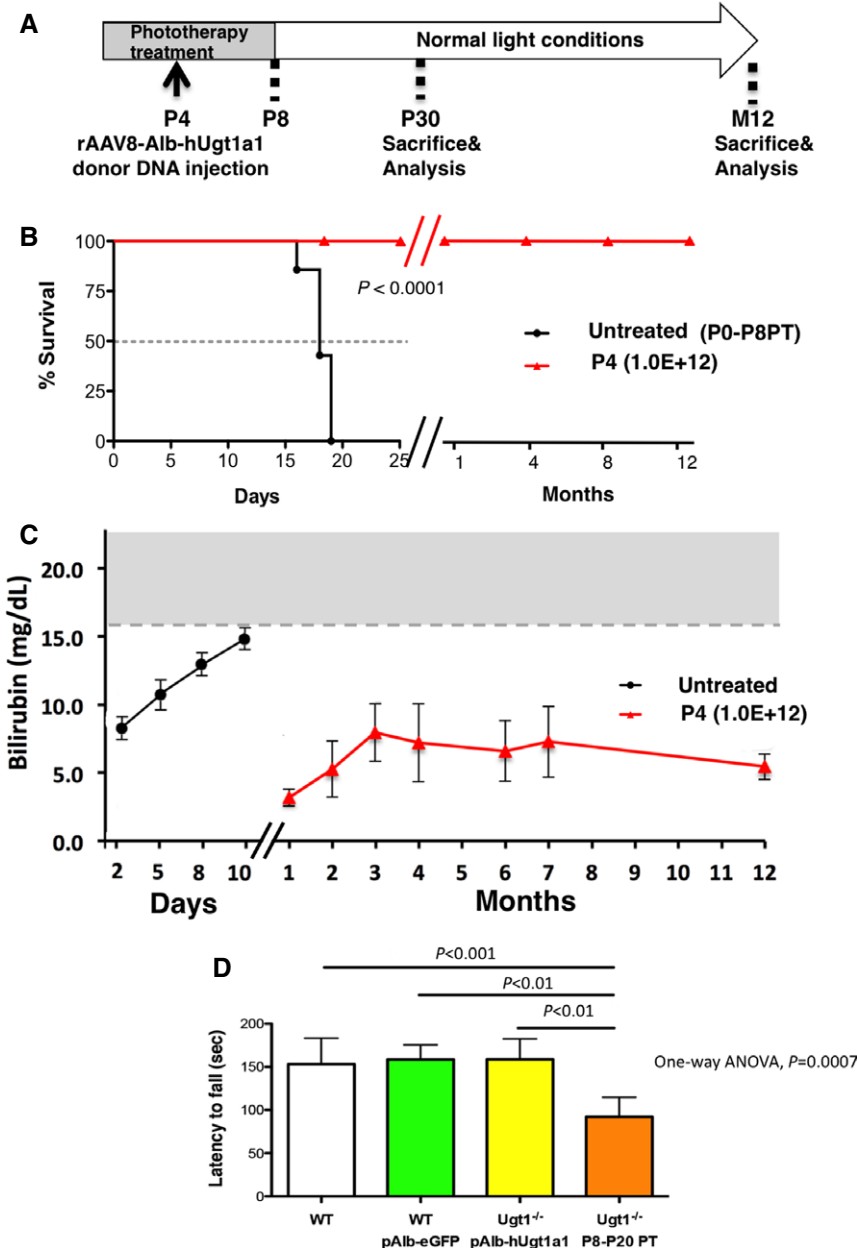

**Figure 2. Transduction of Ugt1$^{-/-}$ mice in lethal conditions with the rAAV-Alb-hUgt1a1 donor vector rescues lethality.**

A   Experimental strategy. Ugt1$^{-/-}$ mutant mice were injected at P4 with rAAV8 pAlb-hUGT1A1 (1.0E12 vgp/mouse) and maintained under PT up to P8.

B   Kaplan–Meier survival curve. All rAAV-treated mutant mice ($n = 5$) survived, while all mutant mice treated only with PT up to P8 died before P19 ($n = 6$). $P \leq 0.0001$, Log-rank (Mantel–Cox) test.

C   Plasma bilirubin was determined up to 12 months. Untreated mice received no PT, and all died before P15. The gray area in the graph indicates the range of TB levels resulting in brain damage and death. Untreated, $n = 3$–6; treated, $n = 5$.

D   Rotarod test of WT animals ($n = 13$) and WT treated with rAAV-Alb-eGFP donor vector (P4, 1.0E12 vgp/mouse, $n = 5$), and Ugt1$^{-/-}$ mice ($n = 5$) as described in panel (A). Ugt1$^{-/-}$ mice temporally treated with PT (from P8 to P20, $n = 3$) with cerebellar abnormalities were used as control (Bortolussi *et al*, 2014a) (Ugt1$^{-/-}$ P8-P20 PT). One-way ANOVA: $P = 0.0007$; Bonferroni *post hoc* tests: WT versus Ugt1$^{-/-}$ P8-P20 PT, $t = 4.612$, $P < 0.001$; WT pAlb-eGFP versus Ugt1$^{-/-}$ P8-P20 PT, $t = 4.381$, $P < 0.01$; Ugt1$^{-/-}$ pAlb-hUGT1A1 vs. Ugt1$^{-/-}$ P8-P20 PT, $t = 4.386$, $P < 0.01$; all others, ns.

Data information: Results are expressed as mean ± SD.

mice (Fig 2D). Mutant mice temporarily treated with PT from P8 to P20 with cerebellar abnormalities due to bilirubin neurotoxicity (Bortolussi *et al*, 2014a), used as positive control for the rotarod test, showed a significant reduction in their performance.

We then performed the molecular analysis of AAV-Alb-eGFP-treated livers. RT–PCR of total liver RNA using primers specific for chimeric cDNA showed the presence of the expected band (Fig 3A), which was confirmed by restriction enzyme digestion, cloning, and

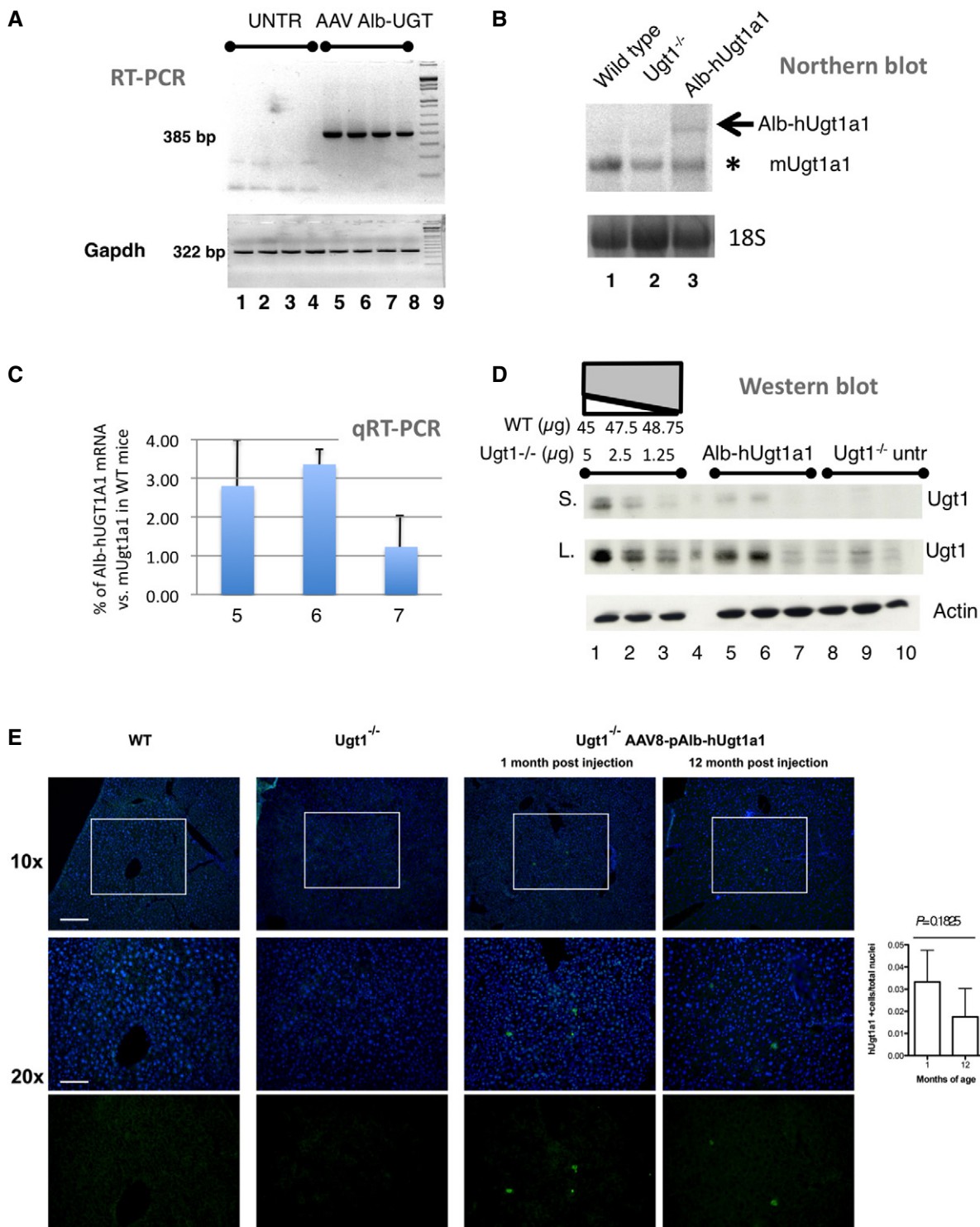

**Figure 3.**

sequencing (Fig EV4A–C). Northern blot analysis showed a specific band of the expected size, corresponding to the chimeric Alb-P2A-hUgt1a1 mRNA (Fig 3B). The levels of the chimeric Alb-hUgt1a1 mRNA in treated mice at P30 were about 2.5% of those present in WT animals (mUgt1a1, Figs 3C and EV4D and E), but decreased in

4- and 12-month-old mice to 1.3 and 0.5%, respectively, as determined by quantitative RT–PCR using primers specific for each mRNA species (Fig EV4F).

Western blot analysis showed the presence of a Ugt1a1-specific band only in AAV8-Alb-hUgt1a1-treated animals (Fig 3D). A rough

◄

**Figure 3.    Molecular and histological analysis of Ugt1$^{-/-}$ mice treated with the rAAV-Alb-hUgt1a1 donor vector.**

A    RT–PCR of liver RNA samples from Fig EV3D and panel (E) (P4 1.0E12 vgp/mouse). *Gapdh* mRNA was used as housekeeping control. UNTR indicates mice treated only with PT.

B    Northern blot of total liver RNA from WT, Ugt1$^{-/-}$ untreated (PT up to P15) or treated with rAAV8-Alb-hUgt1a1 (1E12 vgp/mouse at P4, PT up to P8, sacrificed at P30). Black arrow, Alb-hUgt1a1 mRNA; asterisk, endogenous mUgt1a1 mRNAs.

C    Quantitative RT–PCR of liver total RNA from mutant mice treated with rAAV8-Alb-hUgt1a1 (1E12 vgp/mouse at P4, PT up to P8, sacrificed at P30). Mice were the same of panel (D). Alb-hUgt1a1 mRNA levels are relative to endogenous Ugt1a1 mRNA levels of WT mice. RNA samples were analyzed in duplicate.

D    WB analysis of liver protein extracts (50 μg) from Ugt1$^{-/-}$ mice untreated (only PT up to P15, lanes 8–10) or treated with rAAV8-Alb-hUgt1a1 (lanes 5–7), at 1 month. As control, WT plus Ugt1$^{-/-}$ liver extracts (1.25, 2.5 μg, and 5 μg of WT, completed with Ugt1$^{-/-}$ extract up to 50 μg, corresponding to 2.5, 5.0, and 10.0%, respectively, lanes 1–3). Short and long exposures are shown (S. and L., respectively). Loading control, actin. Estimation of Ugt1a1 levels in treated mice (lanes 5–7) by comparison with the signal obtained in lanes 1–3 (mix of extracts from WT and untreated mutant livers), ratio Ugt1a1/actin (RU, mean ± SD), 5.3 ± 2.1.

E    IF analysis of liver sections from WT and Ugt1$^{-/-}$ mice treated with rAAV8 pAlb-hUgt1a1 at P4 (1.0E12, vgp/mouse, sacrificed at M1 or M12), using a human-specific anti-Ugt1a1 antibody. Nuclei were counterstained with Hoechst. *n* = 3–4 per time point/treatment. Right panel: Quantification of sections at 1 and 12 months. Student's *t*-test, *P* = 0.1825, NS. Scale bars, 10×, 200 μm, 20×, 100 μm.

Data information: Results are expressed as mean ± SD.

estimation of Ugt1a1 levels in treated mice was done by comparing the specific signal obtained in protein extracts of treated mutant mice (50 μg) with that obtained by loading increasing amounts of WT liver extracts (1.25, 2.5, and 5 μg) mixed with extracts from mutant mice, to complete 50 μg of protein load. WB quantification showed that treated mutant mice had about 5–6% of WT protein levels (Fig 3D). We could not detect any specific signal in liver extracts from AAV-Alb-hUgtt1a1-treated 4- and 12-month-old mice, even after long exposure of the membranes (Appendix Fig S1A–C). The anti-Ugt1a1 antibody detected an unspecific band in untreated mutant extracts that co-migrated with the expected band present in WT and mutant-AAV-Alb-hUgt1a1-treated extracts, not allowing the detection of Ugt1a1 in extracts containing less than 2.5% of WT Ugt1a1 (Fig 3D).

Histological analysis showed that liver architecture was normal (Fig EV5A). Livers of 12-month-old treated mice did not show any sign of fibrosis or fatty liver (Masson's trichrome and Oil red O staining, respectively, Fig EV5B and C). However, since we observed increased infiltrates in both untreated and treated Ugt1$^{-/-}$ animals (Fig EV5A), suggesting the presence of inflammation, we analyzed inflammation markers in both Ugt1$^{-/-}$ mice and WT mice treated with the Ugt1a1 and eGFP donor vectors, respectively. Despite the absence of fibrosis or other abnormalities, we observed a trend toward the upregulation of inflammatory markers in mice treated with eGFP and hUgt1a1 donor constructs (Appendix Figs S2 and S3A and B). Immunohistochemical analysis with a human-specific anti-Ugt1a1 Ab revealed the presence of hUgt1a1-positive hepatocytes, both at 1 and 12 months post-injection, although they were less frequent in the 12-month time point (Fig 3E; ~0.032% and 0.015% at 1 and 12 months, respectively), confirming the results obtained by WB.

## Discussion

We successfully applied the promoterless gene targeting without nucleases approach to a severe and clinically relevant lethal mouse model of the CNSI, a liver disease of the newborns. We fully rescued lethality, biochemical, and functional abnormalities by transducing mutant neonate mice with the rAAV8-Alb-hUgt1a1 donor vector. Importantly, our results represent a proof of principle that this therapeutic approach can be also effective for cytoplasmic and ER membrane-bound proteins. If safety and efficacy concerns are refined, this strategy has the potentiality to replace liver transplantation, a challenging and risky procedure with significant shortcomings (Adam *et al*, 2012; Immordino *et al*, 2014).

This methodology, originally applied to a hemophilia B mouse model (Barzel *et al*, 2015), exploits the recombination capacity of rAAV vectors (Russell & Hirata, 1998). Recombination efficiency in the liver reached up to 0.2–0.5% (Barzel *et al*, 2015), a value that is in line with the results obtained in our experiments (0.14% of GFP-positive hepatocytes) with the rAAV8-Alb-eGFP donor construct. Yet, a lower value (0.033%) was observed with the rAAV8-Alb-hUgt1a1 donor construct. This could be related to construct-specific differences in gene targeting rate, or to differences between albumin zonal expression (Poliard *et al*, 1986) and AAV8 transduction pattern (Bell *et al*, 2011; Dane *et al*, 2013), resulting in hUgt1a1 expression levels below the limit of detection of IF, or a combination of both. Anyhow, this relatively low gene targeting frequency resulted in about 5% of WT Ugt1a1 enzyme levels, well evident by a systemic change in plasma bilirubin levels. This remarkable therapeutic effect is obtained by supraphysiological levels of the enzyme, transcribed by the potent albumin promoter, in few gene-targeted hepatocytes. The therapeutic potential of this approach is evident, as 5–10% of activity is sufficient to reduce plasma bilirubin to safe levels in human and mice (Fox *et al*, 1998; Sneitz *et al*, 2010; Bortolussi *et al*, 2014b), and 12–22% of WT total liver mass is required in Gunn rats to completely reverse the metabolic defect (Asonuma *et al*, 1992). Nevertheless, since no selective advantage is present in recombinant or WT hepatocytes, a further increase in the overall therapeutic efficacy of the procedure is desirable and will result in important benefits in the patients' treatment.

The stable and targeted insertion of the transgene obtained by the GeneRide approach avoids the loss of therapeutic DNA by hepatocyte duplication, as observed in gene replacement approaches based on episomal AAV DNA (Cunningham *et al*, 2008; Wang *et al*, 2012; Bortolussi *et al*, 2014b), a critical issue as many liver genetic diseases have pediatric onset. Vector loss in animal models occurs very rapidly after neonatal AAV delivery, directly correlating with liver growth (Cunningham *et al*, 2008; Wang *et al*, 2012). Loss of therapeutic efficacy may require re-administration of the vector, an approach that is at the moment precluded by anti-AAV neutralizing antibodies generated after the first administration (Riviere *et al*, 2006; Murphy *et al*, 2009), preventing enrollment of young patients in conventional AAV-mediated gene therapy trials.

Yet, the current approach presented here requires doses of AAV that are much higher than those used in "conventional" AAV-mediated gene replacement hemophilia B clinical trials (Nathwani *et al*, 2009, 2011, 2014), raising important concerns of AAV systemic and liver toxicity, particularly when administered to neonate/pediatric patients. Therefore, further improvement of the GeneRide strategy is needed to allow its clinical development.

On the other hand, the proposed strategy should limit some concerns present with other gene therapy approaches (Valdmanis *et al*, 2012; Fu *et al*, 2013), such as increased risk of developing HCC by off-target insertion of promoter-driven constructs (Donsante *et al*, 2007; Chandler *et al*, 2015). However, a deeper analysis of potential vector genotoxicity and other potential harmful effects derived for such a high dose should be carried out carefully before considering this methodology safe. Still, the normal liver histology, without signs of fibrosis or fatty liver, and the absence of tumors, 1 year after neonatal viral delivery, provide encouraging safety data supporting the procedure.

A potential increase in overall efficacy, with a consequent reduction in AAV doses, may be obtained by different strategies ranging from more efficient serotypes or codon-optimized hUgt1a1 cDNA versions (Ronzitti *et al*, 2016). Importantly, the recombination efficiency was higher when the AAV administration was performed in multiple doses, potentially opening a new perspective in AAV administration. A smart strategy conferring a selective advantage to the targeted hepatocytes was recently developed (Nygaard *et al*, 2016). However, this approach requires temporary treatment of patients with a hepatotoxic drug, an issue that requires further studies. Other strategies could be based in blocking NHEJ or enhancing HDR by using specific compounds (Paulk *et al*, 2012; Srivastava *et al*, 2012; Maruyama *et al*, 2015). However, the real efficacy of these approaches needs to be tested in the proper animal models. Indeed, the use of engineered nucleases is a promising strategy to increase HR rate and the overall therapeutic efficacy (Anguela *et al*, 2013; Sharma *et al*, 2015; Yang *et al*, 2016), allowing an important reduction in AAV doses to therapeutic and safer levels. However, transient expression of the nucleases may be required to limit most safety issues related to their application.

The presented results rule out the important concern regarding the final fate of proteins encoded by the "bicistronic" mRNAs, with gene products directed to different secretory pathways or subcellular compartments. In fact, no information was available for non-secreted proteins since the approach was previously tested with the secreted coagulation factor IX (Barzel *et al*, 2015). We showed cytoplasmic localization of eGFP and hUgt1a1, which correlated with gene expression levels and with plasma bilirubin levels and rescue of lethality. To note, CNSI patients having truncated Ugt1a1 proteins have no glucuronidation activity (Kadakol *et al*, 2000; Bosma, 2003; Canu *et al*, 2013), due to the rapid degradation of the mislocalized protein with no transmembrane domain (Emi *et al*, 2002).

Importantly, the impact of our findings could extend the knowledge obtained with hemophilia, where the transgene product is secreted, to other metabolic disorders where the gene product is cytoplasmic or internal membrane-bound, in which the disease phenotype is caused by accumulation of a soluble toxic intermediate, such as bilirubin in the CNSI or ammonia in urea cycle disorders. Overall, the success obtained in the present work supports further development of the approach to other genetic diseases.

# Materials and Methods

## Animals

Animals were housed and handled according to institutional guidelines, and experimental procedures approved by International Centre for Genetic Engineering and Biotechnology (ICGEB) review board, with full respect to the EU Directive 2010/63/EU for animal experimentation. Ugt1a mutant mice used in this study were at least 99.8% FVB/NJ genetic background (Bortolussi *et al*, 2014a), obtained after more than 10 backcrosses with FVB/NJ WT mice. Experimental groups were composed of mice of mixed gender. Mice were kept in a temperature-controlled environment with a 12-h/12-h light–dark cycle, with a standard diet and water *ad libitum*.

## Plasmids

To generate the pAlb-hUGT1A1 and pAlb-eGFP donor vectors, the FIX cDNA present in the pAB288 plasmid (Barzel *et al*, 2015) was removed by NheI digestion and a short linker was inserted to allow further cloning of the hUGT1a1 and eGFP cDNAs into the BstX I-NheI sites. For episomal expression, an AAV8-AAT-eGFP vector was used (Bortolussi *et al*, 2014b).

## Production, purification, and characterization of the rAAV vectors

The AAV vectors used in this study are based on AAV type 2 backbone, and infectious vectors were prepared by the AAV Vector Unit at ICGEB Trieste (http://www.icgeb.org/avu-core-facility.html) in HEK293 cells by a cross-packing approach whereby the vector was packaged into AAV capsid 8, as described (Bortolussi *et al*, 2014b).

## Animal treatment

For the AAV gene transfer procedure, pups at different post-natal day (P2, P4, and P10) were intraperitoneally injected with the indicated vectors, at the indicated doses. Juvenile animals (P30) were transduced by retro-orbital injection.

For P2, P4, and P10 ages, multiple injections were performed as follows: the total dose was divided into three injections, separated by a 5-h window. All mutant and WT littermates newborns were exposed to blue fluorescent light (20 $\mu W/cm^2$/nm, Philips TL 20W/52 lamps; Philips, Amsterdam, The Netherlands) for 12 h/days (synchronized with the light period of the light/dark cycle) for the indicated period after birth and then maintained under normal light conditions. Intensity of the blue lamps was monitored monthly with an Olympic Mark II Bili-Meter (Olympic Medical, Port Angeles, WA, USA).

## Phototherapy treatment

Newborn pups were exposed to blue fluorescent light ($\lambda$ = 450; 20 $\mu W/cm^2$/nm; Philips TL 20W/52 lamps; Philips, Amsterdam, The Netherlands) for 12 h/days (synchronized with the light period of the light–dark cycle). Intensity of the lamps was monitored monthly with an Olympic Mark II Bili-Meter (Bortolussi *et al*, 2012).

## Bilirubin determination in plasma

Blood was obtained from anesthetized mice by facial vein bleeding and by cardiac puncture. Plasma was obtained by adding EDTA 200 mM and centrifuging at 400 *g* for 15 min at room temperature. Bilirubin determination in plasma was performed following the instructions of the supplier (BQ KITS) (Bortolussi *et al*, 2014a).

## RNA preparation and RT–PCR

Whole livers were extracted and reduced to powder with a mortar and liquid nitrogen, and stored at −80°C. Total RNA from mouse liver was extracted by using EuroGOLD Trifast (Euroclone) reagent according to manufacturer's instructions. About 1 μg of total RNA was reverse-transcribed using M-MLV (Invitrogen) and oligo-dT primer according to manufacturer's instructions. The resulting cDNA was used to perform either RT–PCR by using specific primers listed in Appendix Table 1, to amplify the hybrid RNA, the endogenous mouse *Albumin* cDNA, or the *Gapdh* housekeeping gene.

## mRNA quantification analysis (qRT–PCR)

qRT–PCR experiments were performed as previously described (Bockor *et al*, 2017; Vodret *et al*, 2017). Briefly, total RNA from mouse livers was prepared using EuroGOLD Trifast (Euroclone, Milano, Italy). One microgram of total RNA was reverse-transcribed (Bockor *et al*, 2017), and 1:10 diluted cDNA (2 μl) was used to perform qPCR using the specific primers listed in Appendix Table 1. qPCR was performed using the iQ SYBR Green Supermix (Bio-Rad) and a C1000 Thermal Cycler CFX96 Real Time System (Bio-Rad). Expression of the gene of interest was normalized to *Gapdh*, for quantification of the inflammatory markers, and to albumin for estimation the relative expression of the transgenes. Data were analyzed using the ΔΔCt method.

## Northern blot analysis

Total RNA from mouse livers was prepared using EuroGOLD Trifast (Euroclone, Milano, Italy). To perform northern blot analysis, 20 μg of total liver RNA was denatured and run on a 1.2% agarose formaldehyde gel and blotted onto a nylon membrane (Hybond-N; Amersham Biosciences, Uppsala, Sweden), as previously described (Bortolussi *et al*, 2012). The membrane was incubated with UltraHyb prehybridization solution (Ambion, Austin, TX, USA) and subsequently hybridized with a $P^{32}$-radiolabeled probe spanning the human Ugt1a1 cDNA (1.6 Kb) obtained by digesting pGG2-hUgt1a1 vector with XhoI and NotI restriction enzymes (Bortolussi *et al*, 2014b). After being washed, the membrane was exposed overnight using a Cyclone phospho-screen (Packard Bioscience Co., Downers Grove, IL, USA) and the detection of the radioactive signal was done with Cyclone Storage phospho-imager (Packard Bioscience). 18S rRNA intensity was used as loading control.

## Preparation of total protein extracts and Western blot analysis

Liver total protein extracts were obtained as described (Bortolussi *et al*, 2012). Briefly, liver powder was homogenized in RIPA buffer (150 mM NaCl, 1% NP-40, 0.5% DOC, 0.1% SDS, 50 mM Tris–HCl, pH 8, and 2× protease inhibitors (Roche), centrifuged at max speed for 10 min at 4°C. Total protein concentration was determined by Bradford (Bio-Rad). For the Western blot analysis, 50 μg of total protein extracts from mutant treated, mutant untreated, and WT animals was analyzed as described previously (Bortolussi *et al*, 2012). Primary antibodies used were as follows: anti-human UGT1 rabbit polyclonal antibody (1:800; H-300, Santa Cruz Biotechnology) and anti-actin (1:2,000; A-2066, Sigma-Aldrich).

## Rotarod analysis

One-month-old mice, Ugt1a$^{-/-}$ treated with rAAV8-Alb-hUGT1A1, WT, WT treated with rAAV8-Alb-eGFP and Ugt1a$^{-/-}$ temporarily treated with PT (from P8 to P20). This latter group (PT-P8-P20) received a temporary PT treatment that rescues of lethality but mice present neurological damage (Bortolussi *et al*, 2014a), and was used as positive control of the test. Mutant mice treated with rAAV8-Alb-hUGT1A1 at P4 (1.0E12 vgp/mouse) received PT treatment from P0 to P8. WT mice treated with rAAV8-Alb-eGFP at P4 (1.0E12 vgp/mouse) were used as controls. All animals were tested for their coordination and balance ability with an accelerating apparatus as previously described (Bortolussi *et al*, 2012).

## Histological analysis

After sacrifice, livers were fixed with 4% PFA in PBS overnight at 4°C and then kept in 20% sucrose in PBS and 0.02% sodium azide at 4°C.

Paraffin-embedded sections (5 μm) were stained with hematoxylin–eosin (H&E) and Masson's trichrome as previously described (Bortolussi *et al*, 2012, 2014b). Oil red staining was performed according to manufacturer instructions (BioOptica, Milano, Italy).

For eGFP experiments and anti-hUgt1a1 immunofluorescence, specimens were frozen in optimal cutting temperature compound (BioOptica, Milano, Italy) and 14-μm slices were obtained in a cryostat: (i) eGFP immunofluorescence liver specimens were stained with Hoechst (10 μg/ml) and mounted with Mowiol 4-88 (Sigma). (ii) hUgt1a1 immunofluorescence liver specimens were incubated with sodium citrate prior to blocking solution step. Next, specimens were blocked with 10% NGS (Dako) and then incubated with the primary antibody for 2 h at RT in 2% NGS blocking solution with anti-hUgt1a1 1:200 (Sigma, St. Louis, MO). After 3 × 5 min washes with blocking solution, specimens were incubated with secondary antibody (Alexa Fluor 488, Invitrogen Carlsbad, CA) for 2 h at RT. Nuclei were visualized by addition of Hoechst (10 μg/ml, Invitrogen) for 5 min after secondary antibody solution.

Images were acquired on a Nikon Eclipse E-800 epi-fluorescent microscope with a charge-coupled device camera (DMX 1200F; Nikon, Amstelveen, The Netherlands). Digital images were collected using ACT-1 (Nikon) software.

Quantification of eGFP-positive cells was performed as follows: Each animal was imaged in four liver sections. Three fields (2 mm × 2 mm) per section (three sections/animal) were analyzed. Each field contained an average of 15,000 nuclei. Measurements were averaged for each animal, and the results were expressed as mean ± SD for each treatment.

## The paper explained

### Problem

The Crigler-Najjar syndrome type I is a rare genetic disease caused by mutations in the Ugt1 gene, the only enzyme able to conjugate bilirubin. It is characterized by severe unconjugated hyperbilirubinemia since birth, with lifelong risk of permanent neurological damage, kernicterus, and death. The only permanent cure for the disease is liver transplantation. Whether *in vivo* gene targeting approach to Ugt1$^{-/-}$ neonate mice is sufficient to reduce bilirubin toxicity and rescue lethality was still not determined.

### Results

We targeted a promoterless hUGT1a1 cDNA into the albumin locus, without the use of nucleases. We showed the complete rescue from neonatal lethality, with a therapeutic reduction in plasma bilirubin lasting for at least 12 months. The transgene was stably inserted into the genome and was active in actively duplicating hepatocytes.

### Impact

Our studies suggest that the promoterless gene targeting strategy without the use of nucleases is a feasible therapeutic strategy for Crigler-Najjar syndrome and could be applied to other liver genetic disorders. However, a more efficient recombination rate may be necessary for its application in the clinical setting.

### Statistics

The Prism package (GraphPad Software, La Jolla, CA) was used to analyze the data. Results are expressed as mean ± SD. Values of $P < 0.05$ were considered statistically significant. Depending on the experimental design, Student's *t*-test or one-way or two-way ANOVA, with Bonferroni's *post hoc* comparison tests, was used, as indicated in the legends to the figures and text. All datasets were analyzed with the Shapiro–Wilk test to assess normal distribution of the data. These results are shown in Appendix Table S2. Kaplan–Meier survival curves were analyzed by the Log-rank (Mantel-Cox) test.

### List of oligonucleotides

See Appendix Table S1.

**Expanded View** for this article is available online.

### Acknowledgements

Thanks to Luka Bočkor for help in the initial phases of the project, and to Willy De Mattia and Stefano Artico for animal husbandry. The authors thank Prof. E. Tongiorgi from the Department of Life Sciences, University of Trieste, Italy, for the microscope facility resources. This work was supported by intramural funds to AFM and NIH R01 HL064274 to MAK.

### Author contributions

FP, GB, ADC, SV, and AI performed the experiments; LZ prepared the AAV vectors; FP, GB, and AFM prepared the Figures; AB designed experiments and provided the GeneRide backbone donor construct; MAK designed experiments, provided vector constructs, and wrote the manuscript; AFM designed and analyzed the experiments, and wrote the manuscript. All authors discussed the results and commented on the manuscript at all stages.

### Conflict of interest

M.A.K. and A.B. are shareholders for LogicBio Therapeutics that has licensed the GeneRide technology from Stanford University.

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
