## [Review Process File · EMBO Molecular Medicine]

Promoterless gene targeting without nucleases rescues lethality of a Crigler-Najjar syndrome mouse model

Fabiola Porro, Giulia Bortolussi, Adi Barzel, Alessia De Caneva, Alessandra Iaconcig, Simone Vodret, Lorena Zentilin, Mark A. Kay and Andrés F. Muro

Corresponding author: Andrés Muro, International Centre for Genetic Engineering and Biotechnology

Review timeline:

Submission date:	24 January 2017
Editorial Decision:	03 March 2017
Revision received:	02 June 2017
Editorial Decision:	13 June 2017
Revision received:	27 June 2017
Accepted:	04 July 2017

Transaction Report:

Editor: Céline Carret

1st Editorial Decision

03 March 2017

Thank you for the submission of your manuscript to EMBO Molecular Medicine. We have now heard back from the three referees whom we asked to evaluate your manuscript. Although the referees find the study to be of potential interest, they also raise a number of concerns that must be addressed in the next final version of your article.

As you will see from the reports below, the referees find the study interesting and well performed. However, referees 2 and 3 are critical of the statistics and exact n and p-values along with better statistical description are needed. Better-described experimental protocols and discussions are also to be provided. Finally and importantly, both referees 1 and 2 request additional experiments to strengthen the conclusions, and those are absolutely necessary for the paper to move forward. You will see that referee 2 suggests shortening the study to a report format to maximize impact and we would strongly encourage you to follow this recommendation.

Please note that it is EMBO Molecular Medicine policy to allow only a single round of major revision and that, as acceptance or rejection of the manuscript will depend on another round of review, your responses should be as complete as possible.

I look forward to receiving your revised manuscript.

***** Reviewer's comments *****

Referee #1 (Remarks):

This paper relates the usage of AAV8 mediated targeted insertion of the UGT1A1 cDNA into the albumin locus in hepatocytes to correct Crigler-Najjar syndrome type 1 in a mouse model. Data are indicative of efficacy in enabling survival at cost of a mild persisting hyperbilirubinemia up to 12 months. As such this is novel and interesting.

There are a few questions that need to be addressed :

. In none of the experimental settings, quantitative data about gene insertion and expression are provided. At best, semi quantitative RTPCR data are shown. These quantitative informations are needed to assess the efficacy of the procedure. Also, it will be important to compare fraction of hepatocytes transduced as observed after 30 days (exp 1) or 12 months (exp 2) to the level of protein expression (~ 5 %) in exp 2. Do cells produce physiological or supraphysiological levels of UGT1A ?

In other words do transduced hepatocytes have a competitive selective advantage overtime or not ?

. In addition, if hepatocytes produce supraphysiological amounts of the enzyme, it will be important to assess potential cell toxicity.

. In this respect, and more broadly speaking , histopathological studies of treated livers are needed.

Additional comments

- p5 lane 16. How many fields were counted (how many events) to determine the percentage of GFP(+) cells ?

- Have there been any attempts to study potential off target insertion events ?

- Figure 4D, lanes 4, 5 and 6 correspond I presume to Wblots of enzyme analysis from individual treated mice.

It would be useful to have access to the individual results of enzyme level determinations from the all 7 successfully treated mice to assess variability/reproducibility of the used strategy.

Referee #2 (Remarks):

The paper by Porro et al describes an elegant approach to obtain homology-directed targeting of a UGT1A1 gene in the albumin locus of neonatal mice, with efficiency sufficient to obtain therapeutic levels of transgene expression in a severe murine mode of the Crigler-Najjar syndrome. The approach is not novel, and has been shown in the past to lead to therapeutic levels of FIX protein in hemophilic mice. The study by Porro et al. shows that the approach can be extended to non-secreted liver enzymes, and provides a quantitative measure of HDR-targeting in the albumin locus by using a GFP transgene. The study is straightforward and well controlled, and is carried out by a competent group of investigators with experience in gene therapy for Crigler-Najjar syndrome. However, it is in general light on data and could be more suitable for publication as a short report

A major problem with the study is the dose of vector used to obtain detectable levels of targeted transgene integration. This is almost 2- to3-log higher than the dose necessary to correct serum bilirubin levels by classical, non-targeted AAV-mediated gene therapy. The authors do not address this issue critically and do not provide a rationale as to why HDR-directed targeting should be superior to non-integrating AAV-mediated gene therapy, besides very general statements on a supposedly inferior risk of insertional mutagenesis due to the absence of a trans-activating element in the vector. The projected therapeutic dose in humans for this approach is orders of magnitude higher than those used in clinical trials for hemophilia B, and could potentially be highly toxic or

even life-threatening in pediatric patients affected by a liver disease, and definitely offset the modest, and unproven in humans, risk of insertional mutagenesis by AAV. A fair assessment of risk-benefit ratios of different gene therapy approaches, and suggestions about how the targeting approach could be improved in the direction of reducing the overall vector dose (different vector backbones? serotypes? use of nucleases?) should be at least discussed in the paper.

A second point is the age-dependence of the targeting efficiency, which is not addressed at all in the paper. Although the severe CN mouse model does not allow addressing this issue in terms of phenotypic correction, the use of a GFP construct should allow the authors to test whether targeting efficiency decreases in steady-state, or at least juvenile, livers compared to neonatal organs in normal animals. This is a crucial issue, since it is unlikely that CN patients will ever be treated by gene therapy in the perinatal period. Steady-state or slow-growing hepatocytes could be significantly less susceptible to HDR-directed targeting, and this should be directly addressed in this study since the reagents are available.

Minor issues

1. The number of animals in each treatment and control groups is never mentioned. The statistical test used to determine significance (asterisks) in the different experiments should be mentioned each time in all figure legends.

2. In Figure 2B it is not clear whether there is statistical significance in the difference between P2 and P4 at the 1E12 dose

Referee #3 (Remarks):

This article reports the remarkable finding that nuclease-free gene editing with an AAV vector can rescue lethality in Crigler-Najjar mice. As such, it represents a very promising, and novel approach for treating human Crigler-Najjar patients. Because nucleases are not used, this significantly increases the safety and clinical applicability of the strategy. It follows on the heels of similar publications demonstrating therapeutic improvements in hemophilic mice and also the cure of *Fah*^{-/-} mice by nuclease-free gene editing in the liver. One surprising aspect of this work is the fact that therapeutic levels of *Ugt1a1* were achieved, even though the gene editing frequency was likely only 0.1-1% of hepatocytes. While the survival data are the strength of the paper, and convincing enough to warrant publication in EMBO Molecular Medicine, some of the other conclusions require further data or better controls to be proved conclusively. These relatively minor issues are listed below.

1. In Figure EV4, the authors conclude that multiple vector injections are better than a single injection, but the statistics are not clear. Was a p value calculated? How many mice were studied in each group? And the legend says that one group had IP injections (I assume intraperitoneal), while the other had a portal vein injection. It seems a more valid conclusion would be that IP injection is better than portal vein injection.

2. In Figure 4C, the control group apparently received no phototherapy, while I assume the treated group did. The authors should clearly state which groups received phototherapy for which days, and both the control and experimental groups should receive the same amount and timing of phototherapy. Otherwise, the authors may just be studying the effects of different phototherapy regimens.

3. In Figure 4D, the authors conclude that the edited mice have 5-6% of WT *Ugt1a1* protein levels. However the Figure is not convincing. It appears that the *Ugt1a1*^{-/-} control mice also have a faint band at the position of *Ugt1a1*, hidden behind a relatively dark smear. However, there should be no *Ugt1a1* protein in these mice. Given that it is very unlikely that the gene editing frequency was 5%, this result is a bit suspect, although overexpression of *Ugt1a1* at the albumin locus presumably plays a role. Ideally, the authors should show a more convincing gel for this important quantification, and show more of the gel so we can see why the background smear is so dark in the *Ugt1a1*^{-/-} mice (are there lots of aberrant bands?).

4. In Figure 4E, the experiment again does not account for different phototherapy treatments in the control and experimental groups. The authors should clearly state which groups received phototherapy for which days, and both the control and experimental groups should receive the same amount and timing of phototherapy.

5. Some of the statistics are poorly described. The authors should clearly state how all p values were calculated, and the sizes of all experimental groups.

6. A better discussion of what the gene editing frequency would be helpful, including how the relatively low editing frequencies could produce such a dramatic change in bilirubin levels. This discussion could explain differences in Ugt1a1 protein levels and Ugt1a1 gene editing levels, as well as special considerations associated with editing a non-secreted protein, and how a strong systemic response can be produced by such a small number of cells containing the corrected Ugt1a1 protein. Finally it would be helpful to know if similar levels would be therapeutic in people. When the authors say protein levels of 5% would be therapeutic, do they mean 5% of all hepatocytes expressing normal levels of Ugt1a1, or any number of hepatocytes expressing a total of 5% of Ugt1a1 per liver? I think these issues would be interesting to the reader.

1st Revision - authors' response

02 June 2017

***** Reviewer's comments *****

Referee #1 (Remarks):

General comment:

This paper relates the usage of AAV8 mediated targeted insertion of the UGT1A1 cDNA into the albumin locus in hepatocytes to correct Crigler-Najjar syndrome type 1 in a mouse model. Data are indicative of efficacy in enabling survival at cost of a mild persisting hyperbilirubinemia up to 12 months. As such this is novel and interesting. There are a few questions that need to be addressed:

Reply to General comment:

We thank the reviewer for his/her comments and suggestions to improve the manuscript.

Comment #1

In none of the experimental settings, quantitative data about gene insertion and expression are provided. At best, semi quantitative RTPCR data are shown. These quantitative information are needed to assess the efficacy of the procedure.

Reply to Comment #1:

We have now performed quantitative RT-PCR of both the eGFP and hUGT1a1 approaches. These data are now included in the manuscript in Fig 1C,D (Alb-eGFP) and Fig 3C (Alb-hUgt1a1), and commented in the text (Page 5, lines 15-17; Page 7, lines 23-25).

Comment #2

Also, it will be important to compare fraction of hepatocytes transduced as observed after 30 days (exp 1) or 12 months (exp 2) to the level of protein expression (~ 5 %) in exp 2.

Reply to Comment #2:

We have performed WB of the liver samples of mice at 30 days, 4 and 12 months. Despite the reduced bilirubin levels, we have not observed any signal in the livers of the 4 and 12 months group, probably due to a reduction in total Ugt1a1 production in the recombinant hepatocytes, below the limit of detection. These data were included in Fig EV6 and described in the text (Page 8, lines 4-6).

We have also performed immunohistochemical analysis of liver sections at 1 and 12 months, using a human UGT1A1-specific antibody (Fig 3E). These experiments confirmed the results obtained with the eGFP construct and showed a lower number of positive cells in the older animals (Fig 3E, Page 8, lines 14-16).

Comment #3

Do cells produce physiological or supraphysiological levels of UGT1A ? In other words do transduced hepatocytes have a competitive selective advantage overtime or not ? In addition, if hepatocytes produce supraphysiological amounts of the enzyme, it will be important to assess potential cell toxicity. . In this respect, and more broadly speaking , histopathological studies of treated livers are needed.

Reply to Comment #3:

Immunostaining of liver sections derived from treated animals (1 month) (Fig 3E) confirmed that most hepatocytes had not recombined, similar to the results obtained with the eGFP donor construct (0.15% of the cells, Fig 1B,C and EV1B). Therefore, we conclude that recombinant cells produce supraphysiological levels of hUgt1a1, in order to obtain the overall levels detected by WB (about 5% of WT levels, Fig 3D), consequent to the high expression of the albumin locus. This conclusion was commented in the text (Page 9, lines 13-17).

Regarding a potential competitive selective advantage of recombinant hepatocytes, our data showed the stabilization of plasma bilirubin levels in the long term, suggesting no advantage of the recombinant hepatocytes, and confirming data of patients and Gunn rats treated with WT hepatocytes. In fact, transplantation of WT hepatocyte performed in Crigler-Najjar patients showed transient engraftment of the transplanted cells (Ambrosino et al, 2005; Fox et al, 1998; Lysy et al, 2008). We have commented this point in the revised discussion section (Pages 9 lines 21-22).

To assess potential cell toxicity, we have performed histopathological analysis of treated animals (H&E, Masson trichrome and red-O oil staining). We have not found damage in the treated livers. The absence of liver parenchyma damage is also observed in Crigler-Najjar and Gilbert syndrome patients, and in Gunn rats (Cornelius & Arias, 1972; Fagioli et al, 2013; Malloy & Lowenstein, 1940)(Fig EV7 and EV8). We observed an increase in the number of infiltrated cells in mutant mice, which was more evident in treated animals. However, this increase was apparently not related to the transgene as it was also observed in eGFP-treated WT mice (Fig EV9). Accordingly, we have also analyzed the levels of inflammation markers by qRT-PCR. These results are shown in Fig EV7 and EV9, and described in Page 8, lines 11-16.

Additional comment #1:

- p5 lane 16. How many fields were counted (how many events) to determine the percentage of GFP(+) cells ?

Reply to Additional Comment #1:

We have included the details of the experiment in the Materials and Methods Section

Additional comment #2

- Have there been any attempts to study potential off target insertion events ?

Reply to Additional Comment #2:

The reviewer raises an important point to demonstrate safety of the procedure. We have not analyzed the potential off target sites yet.

We have not observed tumor development in mice analyzed one year after the treatment. Since this procedure is based in the use of a promoter-less construct without the use of engineered nucleases, we expect less risks of off-target expression and transactivation of genes after unspecific integration. We commented these issues in the Discussion Section (Page 9, lines 24-26 and Page 10, lines 1-9)

Additional comment #3:

- **Figure 4D**, lanes 4, 5 and 6 correspond I presume to Wblots of enzyme analysis from individual treated mice. It would be useful to have access to the individual results of enzyme level determinations from the all 7 successfully treated mice to assess variability/reproducibility of the used strategy.

Reply to Additional Comment #3:

We apologize for the lack of details in the Legend to the Figure 4 in the first version of the manuscript. This experiment was initiated with 7 mice in the mutant group. From this group, two mice were sacrificed at 4 months for analysis. The other 5 mice were sacrificed at 12 months. The groups of 1 and 4 months were then completed with other treated mutant mice.

The Western blot analysis of the individual mice is shown in Fig EV6 and commented in the Results Section (Page 8, lines 6-8).

Referee #2 (Remarks):**General comment:**

The paper by Porro et al describes an elegant approach to obtain homology-directed targeting of a UGT1A1 gene in the albumin locus of neonatal mice, with efficiency sufficient to obtain therapeutic levels of transgene expression in a severe murine mode of the Crigler-Najjar syndrome. The approach is not novel, and has been shown in the past to lead to therapeutic levels of FIX protein in hemophilic mice. The study by Porro et al. shows that the approach can be extended to non-secreted liver enzymes, and provides a quantitative measure of HDR-targeting in the albumin locus by using a GFP transgene. The study is straightforward and well controlled, and is carried out by a competent group of investigators with experience in gene therapy for Crigler-Najjar syndrome. However, it is in general light on data and could be more suitable for publication as a short report

Reply to General Comment:

We thank the reviewer for his/her comments and suggestions to improve the manuscript.

Comment #1:

A major problem with the study is the dose of vector used to obtain detectable levels of targeted transgene integration. This is almost 2- to 3-log higher than the dose necessary to correct serum bilirubin levels by classical, non-targeted AAV-mediated gene therapy. The authors do not address this issue critically and do not provide a rationale as to why HDR-directed targeting should be superior to non-integrating AAV-mediated gene therapy, besides very general statements on a supposedly inferior risk of insertional mutagenesis due to the absence of a trans-activating element in the vector. The projected therapeutic dose in humans for this approach is orders of magnitude higher than those used in clinical trials for hemophilia B, and could potentially be highly toxic or even life-threatening in pediatric patients affected by a liver disease, and definitely offset the modest, and unproven in humans, risk of insertional mutagenesis by AAV. A fair assessment of risk-benefit ratios of different gene therapy approaches, and suggestions about how the targeting approach could be improved in the direction of reducing the overall vector dose (different vector backbones? serotypes? use of nucleases?) should be at least discussed in the paper.

Reply to Comment #1:

The dose used in Natwhani et al 2011, 2014 (Nathwani et al, 2014; Nathwani et al, 2011) for gene replacement approaches mediated by AAV with a tissue-specific promoter (hemophilia B clinical trial) is lower than the one used in the present approach. However, we should keep in mind that those approaches were performed in adult individuals, in which liver growth and vector loss is minimal. AAV-mediated neonatal/juvenile gene transfer in animal models results in important vector loss and reduction of therapeutic efficacy (Bortolussi et al, 2014b; Cunningham et al, 2008; Wang et al, 2012). No information on neonatal/pediatric liver gene transfer in humans is available and there is no reason to suppose that the outcome will be different, and it may require higher vector doses and re-administration of the therapeutic vector (not yet possible due to the generation of

neutralizing antibodies consequent to the first administration). The proposed approach demonstrates the feasibility of this approach for Crigler-Najjar, but its efficiency needs improvement to reach a potential application in patients. As suggested by the Reviewer, the vector dose needs to be reduced to avoid potential toxicity in pediatric patients. We included a paragraph addressing these issues in the discussion section (Page 9, lines 21-23 and Page 10, lines 4-12).

Comment#2

A second point is the age-dependence of the targeting efficiency, which is not addressed at all in the paper. Although the severe CN mouse model does not allow addressing this issue in terms of phenotypic correction, the use of a GFP construct should allow the authors to test whether targeting efficiency decreases in steady-state, or at least juvenile, livers compared to neonatal organs in normal animals. This is a crucial issue, since it is unlikely that CN patients will ever be treated by gene therapy in the perinatal period. Steady-state or slow-growing hepatocytes could be significantly less susceptible to HDR-directed targeting, and this should be directly addressed in this study since the reagents are available.

Reply to Comment #2:

We have performed AAV-gene transfer in animals of different ages (P2, P4, P10 and P30), using the eGFP reporter donor vector, to address the potential differences in gene targeting efficiency. These results are included in the manuscript as new panels of Fig 1 (Fig 1B-D) and in the text (Page 5, lines 9-17)

Minor issues

- 1. The number of animals in each treatment and control groups is never mentioned. The statistical test used to determine significance (asterisks) in the different experiments should be mentioned each time in all figure legends.*

Reply to Minor Comment #1:

We apologize for not having included these data in the first version of the manuscript. The missing information was added to the Legends to the Figures.

- 2. In Figure 2B it is not clear whether there is statistical significance in the difference between P2 and P4 at the 1E12 dose*

Reply to Minor Comment #2:

We have now included the statistical significance of the differences between P2 and P4 at the 1E12 dose (Fig 1C).

Referee #3 (Remarks):

General comment

This article reports the remarkable finding that nuclease-free gene editing with an AAV vector can rescue lethality in Crigler-Najjar mice. As such, it represents a very promising, and novel approach for treating human Crigler-Najjar patients. Because nucleases are not used, this significantly increases the safety and clinical applicability of the strategy. It follows on the heels of similar publications demonstrating therapeutic improvements in hemophilic mice and also the cure of Fah^{-/-} mice by nuclease-free gene editing in the liver. One surprising aspect of this work is the fact that therapeutic levels of Ugt1a1 were achieved, even though the gene editing frequency was likely only 0.1-1% of hepatocytes. While the survival data are the strength of the paper, and convincing enough to warrant publication in EMBO Molecular Medicine, some of the other conclusions require further data or better controls to be proved conclusively. These relatively minor issues are listed below.

Reply to General Comment #1:

We thank the reviewer for his/her comments and suggestions to improve the manuscript.

Comment#1

In Figure EV4, the authors conclude that multiple vector injections are better than a single injection, but the statistics are not clear. Was a p value calculated? How many mice were studied in each group? And the legend says that one group had IP injections (I assume intraperitoneal), while the other had a portal vein injection. It seems a more valid conclusion would be that IP injection is better than portal vein injection.

Reply to Comment #1:

We apologize for the mistake present in the legend to Fig EV4 (it is now Fig EV3). All animals were injected using the intraperitoneal route. The legend was corrected. We also included the statistical analysis of the experiment.

Comment#2

In Figure 4C, the control group apparently received no phototherapy, while I assume the treated group did. The authors should clearly state which groups received phototherapy for which days, and both the control and experimental groups should receive the same amount and timing of phototherapy. Otherwise, the authors may just be studying the effects of different phototherapy regimens.

Reply to Comment #2:

The control group shown in Figure 4C (it is now Fig 2C) received no phototherapy, and it shows the time-dependent increase in plasma bilirubin levels leading to death (50% survival was at P11, with no survivor after P15). During phototherapy treatment bilirubin levels are low (mice receiving PT since birth have 3.3 mg/dL at P8), but suspension of the PT treatment results in plasma bilirubin increase reaching 16.6 mg/dL at P15 (Bortolussi et al, 2014a) with no animal surviving beyond P19 (Fig 2B). We have not determined plasma bilirubin levels in the control group shown in Fig 2B, but we assume that after P15 their levels should had been higher than 16.6 mg/dL, leading to neurological damage and death.

We have now described in detail the phototherapy treatment received by each experimental group.

Comment#3

In Figure 4D, the authors conclude that the edited mice have 5-6% of WT Ugt1a1 protein levels. However the Figure is not convincing. It appears that the Ugt1a1^{-/-} control mice also have a faint band at the position of Ugt1a1, hidden behind a relatively dark smear. However, there should be no Ugt1a1 protein in these mice. Given that it is very unlikely that the gene editing frequency was 5%, this result is a bit suspect, although overexpression of Ugt1a1 at the albumin locus presumably plays a role. Ideally, the authors should show a more convincing gel for this important quantification, and show more of the gel so we can see why the background smear is so dark in the Ugt1a1^{-/-} mice (are there lots of aberrant bands?).

Reply to Comment #3:

The reviewer is correct assuming that mutant animals have no Ugt1a1 protein. In effect, non-sense mutations such as the one present in our mice (and in the Gunn rats), result in the absence of both protein and glucuronidation activity (Bortolussi et al, 2012; Bosma, 2003; Canu et al, 2013; Kadakol et al, 2000). The truncated Ugt1a1 enzyme lacking the transmembrane domain (essential for both anchorage in the ER membrane and retention in the ER) is mislocalized and rapidly degraded by the ERAD–proteasome pathway (Emi et al, 2002).

With the aim of improving the sensitivity of the WB and its quality, we have tested different commercial antibodies (H-300, sc-25847; B-4, sc-271268; Millipore, AB10339; BD, 458411;

LSBio, LS-C334543; ABCAM, ab62600) and experimental conditions. Unfortunately, in mouse tissues most of them resulted in low sensitivity and high background/unspecific bands. Antibodies giving less background were considerably less sensible. We selected the H-300 Ab from Santa Cruz and determined the limit of detection of the WB analysis by mixing decreasing amounts of WT liver with increasing amounts of mutant liver (untreated), maintaining constant the total protein load. This experiment showed that the lower limit of detection with our conditions/antibody was approximately 2.5% of WT (Fig 3D; Page 7, line 26 and Page 8, Lines 1-6).

We have repeated the WB experiment and a better quality WB is now shown (Fig 3D).

Regarding the gene editing frequency mentioned by the Reviewer (“*Given that it is very unlikely that the gene editing frequency was 5%, this result is a bit suspect, although overexpression of Ugt1a1 at the albumin locus presumably plays a role.*”), I would like to mention that the levels of UGT1A1 protein are the result of a limited recombination frequency (in the range of ~0.1-0.2%), while the hUG1A1 cDNA is expressed at high levels by the strong albumin promoter, reaching ~5% of WT levels. We have modified the text to clarify this concept (Page 5, lines 7-8; Page 9, lines 11-17).

Comment#4

In Figure 4E, the experiment again does not account for different phototherapy treatments in the control and experimental groups. The authors should clearly state which groups received phototherapy for which days, and both the control and experimental groups should receive the same amount and timing of phototherapy.

Reply to Comment #4:

We have now detailed in the Legend to the Figure (it is now Figure 2D) the treatments that each experimental group had received.

Comment#5

Some of the statistics are poorly described. The authors should clearly state how all p values were calculated, and the sizes of all experimental groups.

Reply to Comment #5:

We apologize for not having included these data in the first version of the manuscript. We have included the requested information in the Legends to the Figures.

Comment#6

A better discussion of what the gene editing frequency would be helpful, including how the relatively low editing frequencies could produce such a dramatic change in bilirubin levels. This discussion could explain differences in Ugt1a1 protein levels and Ugt1a1 gene editing levels, as well as special considerations associated with editing a non-secreted protein, and how a strong systemic response can be produced by such a small number of cells containing the corrected Ugt1a1 protein. Finally it would be helpful to know if similar levels would be therapeutic in people. When the authors say protein levels of 5% would be therapeutic, do they mean 5% of all hepatocytes expressing normal levels of Ugt1a1, or any number of hepatocytes expressing a total of 5% of Ugt1a1 per liver? I think these issues would be interesting to the reader.

Reply to Comment #6:

We have modified the text to make more evident the fact that very low editing frequencies result in therapeutic hUgt1a1 levels (Page 8, Lines 16-18; Page 9, lines 13-23). Within this paragraph of the Discussion Section, it is also included a comment regarding gene editing of a non-secreted protein, and the strong systemic response observed by editing such a small number of cells (Page 9, lines 13-18; Pages 10, lines 23-27 and Page 11, lines 1-4)

References

- Ambrosino G, Varotto S, Strom SC, Guariso G, Franchin E, Miotto D, Caenazzo L, Basso S, Carraro P, Valente ML et al (2005) Isolated hepatocyte transplantation for Crigler-Najjar syndrome type I. *Cell transplantation* 14: 151-157
- Bortolussi G, Baj G, Vodret S, Viviani G, Bittolo T, Muro AF (2014a) Age-dependent pattern of cerebellar susceptibility to bilirubin neurotoxicity in vivo. *Disease models & mechanisms*
- Bortolussi G, Zentilin L, Baj G, Giraudi P, Bellarosa C, Giacca M, Tiribelli C, Muro AF (2012) Rescue of bilirubin-induced neonatal lethality in a mouse model of Crigler-Najjar syndrome type I by AAV9-mediated gene transfer. *FASEB journal : official publication of the Federation of American Societies for Experimental Biology* 26: 1052-1063
- Bortolussi G, Zentilin L, Vanikova J, Bockor L, Bellarosa C, Mancarella A, Vianello E, Tiribelli C, Giacca M, Vitek L et al (2014b) Life-long correction of hyperbilirubinemia with a neonatal liver-specific AAV-mediated gene transfer in a lethal mouse model of Crigler Najjar Syndrome. *Human gene therapy*
- Bosma PJ (2003) Inherited disorders of bilirubin metabolism. *Journal of hepatology* 38: 107-117
- Canu G, Minucci A, Zuppi C, Capoluongo E (2013) Gilbert and Crigler Najjar syndromes: an update of the UDP-glucuronosyltransferase 1A1 (UGT1A1) gene mutation database. *Blood cells, molecules & diseases* 50: 273-280
- Cornelius CE, Arias IM (1972) Animal model of human disease. Crigler-Najjar Syndrome. Animal model: hereditary nonhemolytic unconjugated hyperbilirubinemia in Gunn rats. *The American journal of pathology* 69: 369-372
- Cunningham SC, Dane AP, Spinoulas A, Logan GJ, Alexander IE (2008) Gene delivery to the juvenile mouse liver using AAV2/8 vectors. *Molecular therapy : the journal of the American Society of Gene Therapy* 16: 1081-1088
- Emi Y, Omura S, Ikushiro S, Iyanagi T (2002) Accelerated degradation of mislocalized UDP-glucuronosyltransferase family 1 (UGT1) proteins in Gunn rat hepatocytes. *Archives of biochemistry and biophysics* 405: 163-169
- Faggioli S, Daina E, D'Antiga L, Colledan M, Remuzzi G (2013) Monogenic diseases that can be cured by liver transplantation. *Journal of hepatology* 59: 595-612
- Fox IJ, Chowdhury JR, Kaufman SS, Goertzen TC, Chowdhury NR, Warkentin PI, Dorko K, Sauter BV, Strom SC (1998) Treatment of the Crigler-Najjar syndrome type I with hepatocyte transplantation. *The New England journal of medicine* 338: 1422-1426
- Kadakol A, Ghosh SS, Sappal BS, Sharma G, Chowdhury JR, Chowdhury NR (2000) Genetic lesions of bilirubin uridine-diphosphoglucuronate glucuronosyltransferase (UGT1A1) causing Crigler-Najjar and Gilbert syndromes: correlation of genotype to phenotype. *Hum Mutat* 16: 297-306
- Lysy PA, Najimi M, Stephenne X, Bourgois A, Smets F, Sokal EM (2008) Liver cell transplantation for Crigler-Najjar syndrome type I: update and perspectives. *World J Gastroenterol* 14: 3464-3470
- Malloy HT, Lowenstein L (1940) Hereditary Jaundice in the Rat. *Canadian Medical Association journal* 42: 122-125
- Nathwani AC, Reiss UM, Tuddenham EG, Rosales C, Chowdhury P, McIntosh J, Della Peruta M, Lheriteau E, Patel N, Raj D et al (2014) Long-term safety and efficacy of factor IX gene therapy in hemophilia B. *The New England journal of medicine* 371: 1994-2004

Nathwani AC, Tuddenham EG, Rangarajan S, Rosales C, McIntosh J, Linch DC, Chowdary P, Riddell A, Pie AJ, Harrington C et al (2011) Adenovirus-associated virus vector-mediated gene transfer in hemophilia B. *The New England journal of medicine* 365: 2357-2365

Wang L, Wang H, Bell P, McMenamin D, Wilson JM (2012) Hepatic gene transfer in neonatal mice by adeno-associated virus serotype 8 vector. *Human gene therapy* 23: 533-539

2nd Editorial Decision

13 June 2017

Thank you for the submission of your revised manuscript to EMBO Molecular Medicine. We have now received the enclosed reports from the referees that were asked to re-assess it. As you will see, while the reviewers are more supportive they still do highlight a few issues that must be addressed for the paper to be accepted. In addition, we would like you to amend the paper following the points listed below to make the paper in a suitable format for EMBO Molecular Medicine.

1) Please address the referees' comments in full. Please provide a letter INCLUDING the reviewer's reports and your detailed responses to their comments (as Word file).

2) Statistics:

We note that the distribution of the data is assumed to be normal, which is why you used parametric tests. However, no normality testing is provided/shown. Given that the number of mice analyzed in each instance is rather small (n=4 to 6), we would like to suggest that you use non-parametric tests instead. You can refer to the excellent review published last year in JAHA for guidelines: <http://jaha.ahajournals.org/content/5/10/e004142>

Please provide exact p-values in legends or figures not a range. Some people found that to keep the figures clear, providing a supplemental table with all exact p-values was preferable. You are welcome to do this if you want to. Please make sure to populate the statistical paragraph according to all the questions asked in the author checklist that you have to fill.

Please submit your revised manuscript within two weeks. I look forward to seeing a revised form of your manuscript as soon as possible.

***** Reviewer's comments *****

Referee #1 (Remarks):

Despite improvement, this revised manuscript still contains two important unsolved issues.
1/ No quantitative data of hUgt1a1 gene detection are provided.
2/ It seems to be inconsistent to claim that "treated mutant mice had about 5-6 % of wt protein levels" (fig. 3D) (p8) and then that "the hU1a1 -specific signal was below the detection limit of the technique (about 2.5 % of wt levels) ! A 5-6 % level should accordingly be detected ! Please, clarify!

Referee #2 (Remarks):

The authors addressed most of the issues raised by this reviewer, although the issue of the projected dose in humans is still unsatisfactorily discussed. The authors should definitely be objective in admitting that the dose predicted by the animal model is unprecedentedly high and would limit the clinical development of their strategy unless the efficiency of the process is improved. Genotoxicity is not the major concern here (discussion, page 10, lines 4-5), but rather systemic and liver toxicity, in a context in which, according to data presented at major gene therapy meetings, conventional AAV gene therapy for Crigler-Najjar is currently in clinical development at a two-log lowered dose. The argument of vector dilution with age in a conventional therapy is well taken, but it's really the only one at the moment, and it should be better discussed in a journal addressing a sophisticated medical audience

******* Reviewer's comments *********Referee #1 (Remarks):**

Despite improvement, this revised manuscript still contains two important unsolved issues.

Comment #1

1/ No quantitative data of hUgt1a1 gene detection are provided.

Reply to Comment #1:

We have now included new quantitative data of the hUGT1A1 gene. We have performed the quantification of liver sections of 1 month and 1 year-old animals treated with 1.0×10^{12} vgp/mouse at P4 (Figure 3E, and Discussion Section, Page 9, lines 13-17). We have also included the qRT-PCR data of the chimeric Alb-hUGT1A1 mRNA levels in 4-months and 1 year old mice (Expanded View Figure EV4F). We have also included in the Legend to Figure 3 the data of quantification of the WB shown in Panel D.

Comment #2

2/ It seems to be inconsistent to claim that "treated mutant mice had about 5-6 % of wt protein levels" (fig. 3D) (p8) and then that " the hU1a1 -specific signal was below the detection limit of the technique (about 2.5 % of wt levels) ! A 5-6 % level should accordingly be detected ! Please, clarify

Reply to Comment #2:

We apologize for the poor clarity of this phrase. Treated mutant mice signal was detected only in 1-month samples, but not in those of 4 and 12 months. We have modified the explanation regarding the detection limit of the method (Page 8, lines 6-10), and the Legend to Figure 3D.

Referee #2 (Remarks):

The authors addressed most of the issues raised by this reviewer, although the issue of the projected dose in humans is still unsatisfactory discussed. The authors should definitely be objective in admitting that the dose predicted by the animal model is unprecedently high and would limit the clinical development of their strategy unless the efficiency of the process is improved. Genotoxicity is not the major concern here (discussion, page 10, lines 4-5), but rather systemic and liver toxicity, in a context in which, according to data presented at major gene therapy meetings, conventional AAV gene therapy for Crigler-Najjar is currently in clinical development at a two-log lower dose. The argument of vector dilution with age in a conventional therapy is well taken, but it's really the only one at the moment, and it should be better discussed in a journal addressing a sophisticated medical audience

Reply to Comment

We have modified the Discussion Section according to the Reviewer's suggestions (Discussion Section, Page 10, lines 4-19; Page 11, lines 6- 7).

Corresponding Author Name: Andres Muro

Manuscript Number: EMM-2017-07601